# Mining metagenomes and metatranscriptomes unveils viruses associated with cutaneous squamous cell carcinoma in hematopoietic stem cell transplant recipients

Jana K. Dickter,[1] Yuqi Zhao,[2] Vishwas Parekh,[3] Huiyan Ma,[4] Badri G. Modi,[5] Wai-Yee Li,[6] Saro H. Armenian,[7] Xiwei Wu,[2] Farah R. Abdulla[5]

**ABSTRACT**   We investigated the presence of viral DNA and RNA in cutaneous squamous cell carcinoma (cSCC) tumor and normal tissues from nine individuals with a history of hematopoietic stem cell transplantation (HCT). Microbiome quantification through DNA and RNA sequencing (RNA-seq) revealed the presence of 18 viruses in both tumor and normal tissues. DNA sequencing (DNA-seq) identified *Torque teno virus*, *Saimiriine herpesvirus 1*, *Merkel cell polyomavirus*, *Human parvovirus B19*, *Human gammaherpesvirus-4*, *Human herpesvirus-6*, and others. RNA-seq revealed additional viruses such *as Tobamovirus*, *Pinus nigra virus*, *Orthohepadnavirus*, *Human papillomavirus-5*, *Human herpesvirus-7*, *Human gammaherpesvirus-4*, *Gammaretrovirus*, and others. Notably, DNA-seq indicated that tumor samples exhibited low levels of *Escherichia virus* in three out of nine subjects and elevated levels of *Human gammaherpesvirus-4* in one subject, while normal samples frequently contained *Gammaretrovirus* and occasionally *Escherichia virus*. A comparative analysis using both DNA- and RNA-seq captured three common viruses: *Abelson murine leukemia virus*, *Murine type C retrovirus*, and *Human gammaherpesvirus-4*. These findings were corroborated by an independent data set, supporting the reliability of the viral detection methods utilized. The study provides insights into the viral landscape in post-HCT patients, emphasizing the need for comprehensive viral monitoring in this vulnerable population.

**IMPORTANCE** This study is important because it explores the potential role of viruses in the development of cSCC in individuals who have undergone allogeneic HCT. cSCC is common in this population, particularly in those with chronic graft-versus-host disease on long-term immunosuppression. By using advanced metagenomic and metatranscriptomic next-generation sequencing, we aimed to identify viral pathogens present in tumor and adjacent normal tissue. The results could lead to targeted preventive or therapeutic interventions for these high-risk people, potentially improving their outcomes and management of cSCC.

**KEYWORDS**   cutaneous squamous cell carcinoma, viral infections, allogeneic hematopoietic stem cell transplant survivors, whole-genome sequencing, next-generation sequencing, mining metagenome and metatranscriptome

Cutaneous squamous cell carcinoma (cSCC) following hematopoietic cell transplantation (HCT) is associated with risk factors like chronic graft-versus-host disease (GVHD), often necessitating prolonged immunosuppression, increasing susceptibility to oncogenic viral infections. Diminished immunological responses against these viruses can lead to prolonged shedding and persistent infections, culminating in cellular dysregulation and neoplastic growth (1–3).

**Peer Reviewer** Gerald A. Capraro, Laboratory Corporation of America Holdings, Burlington, North Carolina, USA

Address correspondence to Jana K. Dickter, jdickter@coh.org.

The authors declare no conflict of interest.

*β-Human papilloma virus* (HPV) is a recognized contributor to cSCC, inducing genetic instability and cellular transformation (4–7). However, identifying the specific viral contributions among various potential carcinogenic factors remains challenging (8). T-cell immunity plays a role in suppressing skin cancer, placing immunocompromised hosts at risk. T cell-directed vaccines targeting cutaneotropic HPVs may be a strategy for this at-risk population. (9).

Other viruses, including *Merkel cell polyomavirus* (MCV) and human gammaherpesvirus-4, have been implicated in immunocompromised hosts with cSCC. Molecular studies have shown these viruses in skin cancer biopsies from such individuals (10–14).

Advanced sequencing techniques, like nextgeneration sequencing (NGS), have elucidated viral associations with cancer (15). This technology has enabled the identification of HPV integration sites, disrupted genes, pathways, and epigenetic alterations contributing to cSCC (16, 17). These techniques also reveal that normal skin contains viruses, including bacteriophages (18, 19), Epstein-Barr virus (EBV), *Herpes simplex virus-1*, *Cytomegalovirus* (CMV), *Human herpesvirus 6* (*HHV-6)A* and *6B*, HPV, and MCV (20).

Metagenomic NGS has emerged as a promising avenue for identifying microorganisms associated with cSCC, especially in HCT recipients who have heightened susceptibility to infections. This approach can uncover previously unknown infectious agents linked to cSCC.

In our pilot study, we used metagenomic and metatranscriptomic NGS to analyze cSCC tissue and adjacent normal tissue from nine HCT survivors, aiming to identify candidate pathogens and illustrate their potential role in cSCC development.

Participants were adults over 18 years old with a history of allogeneic HCT and cSCC diagnosed between 2018 and 2022. Three subjects were identified prospectively in 2022, and 29 retrospectively using International Classification of Diseases, 10th Revision codes. Six retrospective samples were selected based on fresher specimens and sufficient nucleic acid yield. This was a 1:1 matched case-control study. Tumor (case) and adjacent normal (control) tissue samples were collected from nine patients, with microbiome quantification assessed through DNA sequencing (DNA-seq) and RNA sequencing (RNA-seq). Detailed endpoints and measurements are provided in the supplemental materials (see https://zenodo.org/records/15053929).

All viruses are identified at the species level, except for gammaretrovirus and tobamovirus, where DNA-seq and RNA-seq were merged, based on virus taxonomy and sequence similarities (>95%). Gammaretroviruses recovered included murine type C retrovirus, Abelson murine leukemia virus, PreXMRV-1 provirus, Moloney murine leukemia virus, murine osteosarcoma virus, Mus musculus mobilized endogenous polytropic provirus, and spleen focus-forming virus. Tobaviruses recovered included pepper mild mottle virus, tobacco mild green mosaic virus, and tomato brown rugose fruit virus isolate Tom1-Jo.

Eight of the nine subjects were male, with a mean age of 69 (range 53–76). Two had a history of multiple malignancies and underwent both autologous and allogeneic HCT. Seven had a history of GVHD treated with immunosuppressive drugs, and four received chemotherapy, immunotherapy, or monoclonal antibody treatment post-HCT. cSCC was located on the scalp, cheek, hand, thigh, chest, and forearm (Table 1).

DNA-seq and RNA-seq of tumor and normal tissues revealed 9 viruses detected by DNA-seq and 12 viruses by RNA-seq (Fig. 1). DNA-seq revealed two viruses found only in tumor tissue (*Carjivirus communis* and *Human gammaherpesvirus-4*), two viruses found only in normal tissue (*Gammaretrovirus* and *Human parvovirus B19*), and five viruses found in both tumor and normal tissues (*Escherichia virus*, HHV-6, MCV, *Saimiriine herpesvirus 1*, and *Torque teno virus*) (Fig. 1A). RNA-seq revealed three viruses found only in tumor tissue (*Alphaendornavirus*, *Human adenovirus 2*, and *Human gammaherpesvirus-4*), one virus identified in only normal tissue (*Bromovirus*), and eight viruses found in both tumor and normal tissues (*Gammaretrovirus*, *Elicom virus 1*, *Escherichia virus*, *Human herpesvirus-7* [HHV-7], *Human papillomavirus-5*, *Tobamovirus*, *Pinus nigra virus 1*, and *Orthohepadnavirus)* (Fig. 1B).

**TABLE 1** Background characteristics of the nine subjects included in the study[a]

| Patient no. | Age at cSCC diagnosis | Sex | Location of cSCC | Hematologic malignancy and HCT | Time since HCT | GVHD immunosuppression history | Other immunosuppression |
|---|---|---|---|---|---|---|---|
| 1 | 53 | M | Cheek | Aplastic anemia s/p[b] alloHCT | 16 years | GVHD prophylaxis: unknown<br>GVHD: mouth, esophagus, skin; treated with steroids, cyclosporine, MMF, and sirolimus | None |
| 2 | 74 | M | Cheek | Polycythemia vera, myelofibrosis s/p alloHCT | 14 months | GVHD prophylaxis: tacrolimus, sirolimus, and ruxolitinib<br>GVHD: skin; treated with steroids and ruxolitinib | Immune complexmediated glomerulopathy s/p rituximab |
| 3 | 65 | M | Dorsum of hand | MDS with excess blasts s/p alloHCT | 26 months | GVHD prophylaxis: tacrolimus and sirolimus<br>GVHD: GI tract, treated with steroids | Maintenance chemotherapy post-HCT with decitabine/venetoclax |
| 4 | 75 | M | Thigh | Mantle cell NHL s/p autoHCT, then developed MDS s/p alloHCT | 10 years | GVHD prophylaxis: tacrolimus and sirolimus<br>GVHD: skin; treated with steroids, rituximab, and MMF | Head and neck squamous cell carcinoma s/p resection, radiation, pembrolizumab |
| 5 | 74 | M | Scalp | AML s/p alloHCT | 12 years | GVHD prophylaxis: tacrolimus and sirolimus<br>GVHD: none | None |
| 6 | 68 | M | Scalp | Mantle cell NHL s/p autoHCT, treated with rituximab maintenance, then developed MDS transformed to AML s/p alloHCT | 10 years | GVHD prophylaxis: tacrolimus, sirolimus, and low-dose methotrexate<br>GVHD: eyes, sclerodermatous; treated with MMF, steroids, and sirolimus | None |
| 7 | 76 | M | Mid-chest | MDS s/p alloHCT | 6 months | GVHD prophylaxis: tacrolimus and sirolimus<br>GVHD: skin; treated with steroids and tacrolimus | None |
| 8 | 70 | M | Forearm | AML s/p alloHCT, then developed DLBCL s/p rituximab, bendamustine, MR-CHOP, and then IEC | 6 years | GVHD prophylaxis: tacrolimus and sirolimus<br>GVHD: skin and GI tract, treated with steroids | ITP treated with rituximab |
| 9 | 60 | F | Forearm | AML s/p alloHCT | 2 years | GVHD prophylaxis: tacrolimus and MMF<br>GVHD: none | None |

[a]alloHCT, allogeneic hematopoietic stem cell transplant; AML, acute myelogenous leukemia; autoHCT, autologous hematopoietic stem cell transplant; DLCBL, diffuse large B-cell lymphoma; GI, gastrointestinal; GVHD, graft-versus-host disease; ITP, immune thrombocytopenia; MDS, myelodysplastic syndrome; MMF, mycophenolate mofetil; NHL, non-Hodgkin lymphoma.
[b]s/p, status post.

By combining DNA- and RNA-seq results, we found that tumor samples had low levels of *Escherichia* virus in three out of nine subjects and elevated levels of *Human gammaherpesvirus-4* in one subject. In contrast, normal samples contained *Gammaretrovirus* in six out of nine subjects, while two out of nine subjects had low levels of *Escherichia* virus. Notably, *Human gammaherpesvirus-4* was absent in all normal samples. DNA-seq and RNA-seq strategies identified three common viruses, including *Gammaretroviruses* (*Abelson murine leukemia virus* and *Murine type C retrovirus*) and *Human gammaherpesvirus-4* (Fig. 1C). Furthermore, the viruses detected by RNA-seq were supported by an independent data set (Fig. 1D; see Fig. S2 at https://zenodo.org/records/15053929).

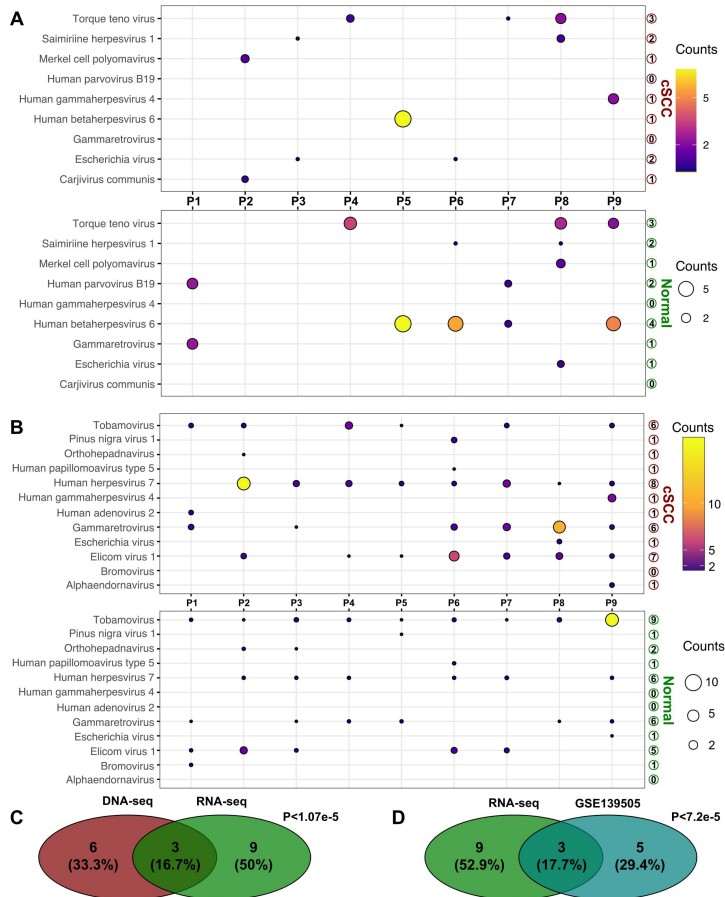

**FIG 1** Viruses detected in the metagenome and metatranscriptome of cSCC patients. (A) Viruses identified through DNA sequencing. (B) Viruses identified through RNA sequencing. (C) Overlapping viruses detected by both DNA-seq and RNA-seq. (D) Overlaps between independent RNA-seq data sets of cSCC with Fisher's exact test.

The finding that *Human gammaherpesvirus-4* was exclusively identified in tumor tissue is noteworthy. EBV-specific expression has been discovered in heart transplant recipients who developed cSCC (11). As EBV is a known oncogenic herpesvirus associated with lymphocytic and epithelial cancers and has been associated with post-transplant malignancies, further research into its role in cSCC is warranted.

RNA-seq identified *Elicom* virus in both normal and tumor tissues in multiple samples. It is a member of the Herpesviridae family, closely associated with swiper virus, which was discovered in female fox fecal samples, but has been reported to have the bivalve mollusk as its host (21). HHV-7, found in both tumor and normal tissues, has been linked to dermatologic conditions and various complications in immunocompromised hosts. The role of HHV-7 in disease, whether as an exogenous antigen in immune reactivations or if viral reactivation is unrelated as a side reaction to the disease, remains unknown (22). *Saimiriine herpesvirus-1*, found in both tumor and normal tissues, is a squirrel monkey alpha herpesvirus with no confirmed cases of human infection (23).

Bacteriophages like *Escherichia virus* and *Carjivirus communis* are prevalent in the gut and may reflect colonization rather than active infection. *Torque teno virus* and *Gammaretroviruses* are integral parts of the human virome (24), with *Gammaretroviruses* making up a notable portion of the human genome, possibly integrated over millions of years (25).

Discrepancies between DNA-seq and RNA-seq results may stem from differences in the biological states of the virus and the sensitivity of each technique. HHV-6, for

example, was detected in DNA-seq in both tumor and normal tissues but not in RNA-seq, likely due to its latent infection state, where its genome is present but not actively transcribed. DNA-seq is more sensitive to detecting stable, integrated, or episomal DNA, while RNA-seq detects only actively transcribed viral genes. If the virus is present at low levels or not replicating, viral transcripts may fall below the RNA-seq detection threshold. The RNA-seq library preparation step, which involves poly(A) selection, could exclude non-polyadenylated viral RNAs, reducing detection sensitivity. Furthermore, viral presence and transcriptional activity can vary across different cell types. DNA-seq detects viral DNA integrated into host cells or present extracellularly, whereas RNA-seq may miss viral transcripts if only a subset of cells expresses the virus at the time of sampling. Detecting viral DNA without RNA suggests a non-productive infection that could still contribute to cellular transformation or immune modulation. The distinction is essential when interpreting the potential role of the virus in tumorigenesis.

This study's small sample size limits its statistical power and ability to detect viral differences between normal and tumor tissues, especially in the context of GVHD. As there are known sex-based predilections for head and neck squamous cell carcinoma, a larger cohort with more female subjects could help identify sex-based differences in viral load and their potential impact on cSCC. This pilot study allowed us to evaluate the feasibility of our design and methods. Larger cohorts are needed to pinpoint specific viruses associated with cSCC in HCT recipients. Once identified, further research is needed to uncover the oncogenic mechanisms through which these viruses contribute to cancer development.

## ACKNOWLEDGMENTS

The authors thank the Department of Pathology at City of Hope National Medical Center, Dr. Stanley Hamilton, and the Shapiro Laboratory for assistance in preparing the samples, and the Integrative Genomics Core at City of Hope National Medical Center for performing sequencing analysis and data collection.BGM has served on the advisory board and speakers' bureau for Regeneron, and has served on the advisory board for Merck.

This work was supported by the City of Hope Comprehensive Cancer Center Department of Medicine Seed Grant (J.K.D.). Work performed in the Integrative Genomics Cores was supported by the National Cancer Institute of the National Institutes of Health under grant number P30CA033572 (Y.Z. and X.W.). The content is solely the responsibility of the authors and does not represent the official views of the National Institutes of Health.

BGM has served on the advisory board and speakers' bureau for Regeneron, and has served on the advisory board for Merck.

## AUTHOR AFFILIATIONS

[1]Department of Medicine, Division of Infectious Diseases, City of Hope National Medical Center, Duarte, California, USA

[2]Beckman Research Institute, City of Hope National Medical Center, Duarte, California, USA

[3]Department of Pathology, City of Hope National Medical Center, Duarte, California, USA

[4]Department of Computational and Quantitative Medicine, City of Hope National Medical Center, Duarte, California, USA

[5]Department of Surgery, Division of Dermatology, City of Hope National Medical Center, Duarte, California, USA

[6]Department of Surgery, Division of Plastic Surgery, City of Hope National Medical Center, Duarte, California, USA

[7]Department of Population Sciences, Department of Pediatrics, City of Hope National Medical Center, Duarte, California, USA

## PRESENT ADDRESS

Vishwas Parekh, Department of Dermatopathology, Diagnostic Pathology Medical Group, Sacramento, California, USA

Farah R. Abdulla, Caris Precision Oncology Alliance, Caris Life Sciences, Irving, Texas, USA

## AUTHOR ORCIDs

Jana K. Dickter  http://orcid.org/0000-0001-5579-1573

## AUTHOR CONTRIBUTIONS

Jana K. Dickter, Conceptualization, Data curation, Formal analysis, Funding acquisition, Investigation, Methodology, Project administration, Supervision, Visualization, Writing – original draft, Writing – review and editing | Yuqi Zhao, Conceptualization, Data curation, Formal analysis, Funding acquisition, Methodology, Project administration, Supervision, Validation, Visualization, Writing – original draft, Writing – review and editing | Vishwas Parekh, Conceptualization, Data curation, Formal analysis, Investigation, Methodology, Resources, Supervision, Validation, Writing – review and editing | Huiyan Ma, Conceptualization, Formal analysis, Methodology, Writing – review and editing | Badri G. Modi, Investigation, Resources, Writing – review and editing | Wai-Yee Li, Writing – review and editing | Saro H. Armenian, Writing – review and editing | Xiwei Wu, Conceptualization, Funding acquisition, Investigation, Methodology, Resources, Supervision, Validation, Writing – review and editing | Farah R. Abdulla, Conceptualization, Supervision, Writing – review and editing

## ADDITIONAL FILES

The following material is available online.

### Open Peer Review

**PEER REVIEW HISTORY (review-history.pdf).** An accounting of the reviewer comments and feedback.

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
