## [Reviewer comments · Microbiology Spectrum]

Microbiology Spectrum

Mining metagenome and metatranscriptome unveils viruses associated with cutaneous squamous cell carcinoma in hematopoietic stem cell transplant recipients

Jana Dickter, Yuqi Zhao, Vishwas Parekh, Huiywan Ma, Badri Modi, Wai-Yee Li, Saro Armenian, Xiwei Wu, and Farah Abdulla

Corresponding Author(s): Jana Dickter, City of Hope

Review Timeline:

Submission Date:	January 7, 2025
Editorial Decision:	February 11, 2025
Revision Received:	February 27, 2025
Accepted:	March 7, 2025

Editor: Hyun Jin Kwun

Reviewer(s): Disclosure of reviewer identity is with reference to reviewer comments included in decision letter(s). The following individuals involved in review of your submission have agreed to reveal their identity: Gerald A Capraro (Reviewer #1)

Transaction Report:

DOI: <https://doi.org/10.1128/spectrum.00053-25>

Re: Spectrum00053-25 (Mining metagenome and metatranscriptome unveils viruses associated with cutaneous squamous cell carcinoma in hematopoietic stem cell transplant recipients)

Dear Dr. Dickter:

Thank you for the privilege of reviewing your work. Below you will find my comments, instructions from the Spectrum editorial office, and the reviewer comments.

The authors should consider submitting the revised article in 'Observations' format, as suggested by the reviewer. To find out about formatting guidelines, refer to the webpage below.
<https://journals.asm.org/journal/spectrum/article-types>

Revision Guidelines

Sincerely,
Hyun Jin Kwun
Editor
Microbiology Spectrum

Reviewer #1 (Comments for the Author):

Thank you for the opportunity to review manuscript Spectrum00053-25, entitled, "Mining metagenome and metatranscriptome unveils viruses associated with cutaneous squamous cell carcinoma in hematopoietic stem cell transplant recipients," by Dickter

et al. This manuscript describes a small study of tumor and normal tissues to detect the presence of oncogenic viruses. Please see my comments below:

Major Comments:

1. This study includes a small number of enrolled patients. The conclusions and stated significance would be strengthened by a more appropriately powered data set.

Minor Comments:

1. Page 8, the first sentence at the top of the page: the word "next" is missing from the introduction of next generation sequencing.

2. Page 10: there are two times in which Escherichia virus is misspelled as Escheriavirus.

Reviewer #2 (Public repository details (Required)):

The authors conducted DNA and RNA sequencing on patient tissue.

Reviewer #2 (Comments for the Author):

The authors of this manuscript conducted a pilot study investigating the presence of various viruses, some oncogenic viruses, that are present in cutaneous squamous cell carcinoma (cSCC) tumor tissue samples from 9 patients who had undergone hematopoietic stem cell transplants (HCT). The authors aim to discern if the presence of the viruses could have a potential role in cSCC tumorigenesis due to long term immunosuppression of HCT patients. Since there is no conclusive causation for the development of cSCC in HCT patients, the idea that oncogenic viruses might replicate more in immunocompromised patients that then lead to the oncogenesis of cSCC is quite novel. The authors utilize next-generation DNA and RNA sequencing to identify viruses found in cSCC and adjacent normal tissue. NGS revealed that human gammaherpesvirus-4, Escherichia virus, and gammaretroviruses were found in cSCC tumor tissue. Notably, human gammaherpesvirus-4 was found only in cSCC tissue, not normal tissue, in one patient. However, due to the limited sample size of this study, there are difficulties in concluding any results from this paper.

1. Due to the very limited scope of this research article (9 patients, 1 Main Figure), I would suggest that this paper be submitted as an observation article. Observation articles provide broad impact to the microbiology field and can present new microbe-disease associations, which this paper entails. Observation articles published from Microbiology Spectrum follow a very similar format to this paper (usually 1 main figure, more observational results/data). See this paper for an example of a similar published observation paper.

2. In the introduction, Epstein-Barr Virus should also be written out as human gammaherpesvirus-4 for clarity.

3. Why do the authors think that DNA and RNA sequencing results did not always match in detection of the virus? For example, DNA viruses such as betaherpesvirus 6 is found in high counts in both normal and tumor tissue but is not detected via RNA sequencing. Does this mean that the virus is present in the tissue, but not expressing viral transcripts? Is one of these sequencing techniques more sensitive than the other?

4. There are sex-based predilections for the occurrence of head and neck SCC, as SCC affects males more than females. Therefore, it should be noted in the discussion to also include more females in the study as well as there could be differences in viral load in the tumors. It is interesting to note that the only patient that has detectable EBV only in the tumor samples by both DNA/RNA sequencing is the only female in the study. Therefore, a large-scale study could reveal sex-based differences in viral load that could cause SCC and explain why males and females have different proclivities for SCC tumorigenesis.

5. Do the authors know the location of the tumors? If so, it would be helpful to list the location in Table 1. Males have been shown to be disproportionately affected by SCC on the eyelid, lips, ear, whereas females often have SCC on cheeks/chin compared to males. Having this information would help shed the light of sex-based disparities and if certain tumor locations are more prone to be infected by certain viruses.

6. If the authors want to keep this manuscript as a research article, more work should be done to strengthen their results. Immunohistochemistry should be conducted to verify the presence of the virus, at least for patient 9 who had detectable EBV.

Response to Reviewers

Reviewer #1 (Comments for the Author):

Thank you for the opportunity to review manuscript Spectrum00053-25, entitled, "Mining metagenome and metatranscriptome unveils viruses associated with cutaneous squamous cell carcinoma in hematopoietic stem cell transplant recipients," by Dickter et al. This manuscript describes a small study of tumor and normal tissues to detect the presence of oncogenic viruses.

Please see my comments below:

Major Comments:

1. This study includes a small number of enrolled patients. The conclusions and stated significance would be strengthened by a more appropriately powered data set.

Response: Changed article type to Observations.

Minor Comments:

1. Page 8, the first sentence at the top of the page: the word "next" is missing from the introduction of next generation sequencing.

Response: corrected

2. Page 10: there are two times in which Escherichia virus is misspelled as Escheriavirus.

Response: corrected

Reviewer #2 (Public repository details (Required)):

The authors conducted DNA and RNA sequencing on patient tissue.

No response required.

Reviewer #2 (Comments for the Author):

The authors of this manuscript conducted a pilot study investigating the presence of various viruses, some oncogenic viruses, that are present in cutaneous squamous cell carcinoma (cSCC) tumor tissue samples from 9 patients who had undergone hematopoietic stem cell transplants (HCT). The authors aim to discern if the presence of the viruses could have a potential role in cSCC tumorigenesis due to long term immunosuppression of HCT patients. Since there is no conclusive causation for the development of cSCC in HCT patients, the idea that oncogenic viruses might replicate more in immunocompromised patients that then lead to the oncogenesis of cSCC is quite novel. The authors utilize next-generation DNA and RNA sequencing to identify viruses found in cSCC and adjacent normal tissue. NGS revealed that human gammaherpesvirus-4, Escherichia virus, and gammaretroviruses were found in cSCC tumor tissue. Notably, human gammaherpesvirus-4 was found only in cSCC tissue, not normal tissue, in one patient. However, due to the limited sample size of this study, there are difficulties in concluding any results from this paper.

1. Due to the very limited scope of this research article (9 patients, 1 Main Figure), I would suggest that this paper be submitted as an observation article. Observation articles provide broad impact to the microbiology field and can present new microbe-disease associations, which this paper entails. Observation articles published from Microbiology Spectrum follow a very similar format to this paper (usually 1 main figure, more observational results/data).

See this paper for an example of a similar published observation paper.

Response: Changed article type to Observations.

2. In the introduction, Epstein-Barr Virus should also be written out as human gammaherpesvirus-4 for clarity.

Response: Corrected.

3. Why do the authors think that DNA and RNA sequencing results did not always match in detection of the virus? For example, DNA viruses such as betaherpesvirus 6 is found in high counts in both normal and tumor tissue but is not detected via RNA sequencing. Does this mean that the virus is present in the tissue, but not expressing viral transcripts? Is one of these sequencing techniques more sensitive than the other?

Response:

The discrepancy between DNA and RNA sequencing results in detecting viruses such as betaherpesvirus 6 (HHV-6) likely arises from differences in the biological states of the virus and the sensitivity of each sequencing technique. Several factors may contribute to this observation:

1) Viral Latency and Transcriptional Activity:

HHV-6 is known to establish latent infections in host tissues, where its viral genome is present as DNA but exhibits little to no transcriptional activity. During latency, viral DNA can be detected through DNA sequencing, but RNA sequencing may not capture viral transcripts due to their low

or absent expression. This explains why the virus is detected in both normal and tumor tissue at the DNA level, yet RNA sequencing yields no evidence of active transcription.

2) Differences in Sensitivity and Detection Limits:

DNA sequencing is inherently more sensitive to detecting the presence of viral genomes, as it captures stable, integrated, or episomal DNA. In contrast, RNA sequencing detects only actively transcribed viral genes. If the virus is present in low abundance or not undergoing active replication, viral transcripts may fall below the detection threshold of RNA sequencing. Additionally, the poly(A) selection step in RNA-seq library preparation may exclude non-polyadenylated viral RNAs, further reducing detection sensitivity.

3) Tissue Heterogeneity and Sampling Variability:

The presence and transcriptional activity of viruses can vary across different cell types within a tissue sample. While DNA sequencing can identify viral DNA integrated into host cells or present extracellularly, RNA sequencing may miss transcripts if only a subset of cells actively expresses the virus at the time of sampling.

4) Biological Relevance:

The detection of viral DNA without corresponding RNA expression suggests a latent or non-productive infection that may still contribute to cellular transformation or immune modulation. This distinction is essential when interpreting the potential role of viruses in tumorigenesis.

In summary, the observed discrepancy is likely due to HHV-6 latency, where viral DNA persists without active transcription. DNA sequencing is more sensitive for detecting latent viruses, while RNA sequencing reflects transcriptional activity, providing complementary insights into the virus's biological state within tissue samples.

4. There are sex-based predilections for the occurrence of head and neck SCC, as SCC affects males more than females. Therefore, it should be noted in the discussion to also include more females in the study as well as there could be differences in viral load in the tumors. It is interesting to note that the only patient that has detectable EBV only in the tumor samples by both DNA/RNA sequencing is the only female in the study. Therefore, a large-scale study could reveal sex-based differences in viral load that could cause SCC and explain why males and females have different proclivities for SCC tumorigenesis.

Response: Added to discussion.

5. Do the authors know the location of the tumors? If so, it would be helpful to list the location in Table 1. Males have been shown to be disproportionately affected by SCC on the eyelid, lips, ear, whereas females often have SCC on cheeks/chin compared to males. Having this information would help shed the light of sex-based disparities and if certain tumor locations are more prone to be infected by certain viruses.

Response: added to table 1.

6. If the authors want to keep this manuscript as a research article, more work should be done to strengthen their results. Immunohistochemistry should be conducted to verify the presence of the virus, at least for patient 9 who had detectable EBV.

Response: changed to Observations

Re: Spectrum00053-25R1 (Mining metagenome and metatranscriptome unveils viruses associated with cutaneous squamous cell carcinoma in hematopoietic stem cell transplant recipients)

Dear Dr. Jana Kubrin Dickter:

Your manuscript has been accepted, and I am forwarding it to the ASM production staff for publication. Your paper will first be checked to make sure all elements meet the technical requirements. ASM staff will contact you if anything needs to be revised before copyediting and production can begin. Otherwise, you will be notified when your proofs are ready to be viewed.

Data Availability: ASM policy requires that data be available to the public upon online posting of the article. DNA-Seq and RNA-Seq data generated during this study should be deposited in a public repository before publication, so please verify all links to sequence records, and make sure that each number retrieves the full record of the data. If a new accession number is not linked or a link is broken, provide production staff with the correct URL for the record. If the accession numbers for new data are not publicly accessible before the expected online posting of the article, publication may be delayed; please contact ASM production staff immediately with the expected release date.

Sincerely,
Hyun Jin Kwun
Editor
Microbiology Spectrum

Reviewer #2 (Comments for the Author):

The authors have done a good job at restructuring the manuscript as an observation paper.

Table 1 Patient 6: rituximab has an extra x in the spelling

On the merged file, Figure 1 is blurry and should be replaced by a higher quality image.